# Peripheral Biomarkers in DSM-5 Anxiety Disorders: An Updated Overview

**DOI:** 10.3390/brainsci10080564

**Published:** 2020-08-17

**Authors:** Matteo Vismara, Nicolaja Girone, Giovanna Cirnigliaro, Federica Fasciana, Simone Vanzetto, Luca Ferrara, Alberto Priori, Claudio D’Addario, Caterina Viganò, Bernardo Dell’Osso

**Affiliations:** 1Department of Mental Health, Department of Biomedical and Clinical Sciences “Luigi Sacco”, University of Milan, 20157 Milan, Italy; nicolajagirone@gmail.com (N.G.); gio.cirnigliaro@gmail.com (G.C.); federica.fasciana@gmail.com (F.F.); simone.vanzetto@unimi.it (S.V.); luca.ferrara@unimi.it (L.F.); caterina.vigano@unimi.it (C.V.); bernardo.dellosso@unimi.it (B.D.); 2Department of Health Sciences, Aldo Ravelli Center for Neurotechnology and Brain Therapeutic, University of Milan, 20142 Milan, Italy; alberto.priori@unimi.it; 3Faculty of Bioscience and Technology for Food, Agriculture and Environment, University of Teramo, 64100 Teramo, Italy; cdaddario@unite.it; 4Department of Clinical Neuroscience, Karolinska Institutet, 17177 Stockholm, Sweden; 5Department of Psychiatry and Behavioral Sciences, Bipolar Disorders Clinic, Stanford University, Stanford, CA 94305, USA; 6“Centro per lo studio dei meccanismi molecolari alla base delle patologie neuro-psico-geriatriche”, University of Milan, 20100 Milan, Italy

**Keywords:** biomarkers, anxiety disorders, peripheral biomarkers

## Abstract

Anxiety disorders are prevalent and highly disabling mental disorders. In recent years, intensive efforts focused on the search for potential neuroimaging, genetic, and peripheral biomarkers in order to better understand the pathophysiology of these disorders, support their diagnosis, and characterize the treatment response. Of note, peripheral blood biomarkers, as surrogates for the central nervous system, represent a promising instrument to characterize psychiatric disorders, although their role has not been extensively applied to clinical practice. In this report, the state of the art on peripheral biomarkers of DSM-5 (Diagnostic and Statistical Manual of Mental Disorders, 5th edition) Anxiety Disorders is presented, in order to examine their role in the pathogenesis of these conditions and their potential application for diagnosis and treatment. Available data on the cerebrospinal fluid and blood-based biomarkers related to neurotransmitters, neuropeptides, the hypothalamic–pituitary–adrenal axis, neurotrophic factors, and the inflammation and immune system are reviewed. Despite the wide scientific literature and the promising results in the field, only a few of the proposed peripheral biomarkers have been defined as a specific diagnostic instrument or have been identified as a guide in the treatment response to DSM-5 Anxiety Disorders. Therefore, further investigations are needed to provide new biological insights into the pathogenesis of anxiety disorders, to help in their diagnosis, and to tailor a treatment.

## 1. Introduction

Anxiety disorders (ADs) are highly prevalent in the general population, often comorbid with psychiatric disorders and medical conditions [1,2], and associated with a negative impact on quality of life and a significant individual and economic burden [3]. As for other psychiatric disorders, the diagnosis is mainly based on clinical examination and psychiatric history, with difficulties in establishing the correct differential diagnosis. Therapeutic strategies, such as antidepressant medications or benzodiazepines (BDZ), are available, although treatment outcomes are only partially satisfactory, with at least one third of patients with ADs not adequately responding to the guidelines-recommended pharmacological treatments [4,5,6,7].

In modern precision medicine, a biomarker has several potential advantages, including predicting disease probability in a pre-clinical stage, helping in differential diagnosis, defining the disease progression and prognosis, and predicting treatment outcomes [5,8]. Indeed, biomarkers emerged as essential instruments in several clinical fields, in particular in cancer and cardiovascular medicine, guaranteeing an increased precision and individualization of diagnostic measures and treatments [8,9]. Translating this approach to psychiatry seems a promising strategy. However, compared to other disciplines, the use of biomarkers in psychiatry is associated with specific challenges, the first one being the limited accessibility to the central nervous system (CNS) in living patients. This is one of the reasons why extensive research focused on peripheral biomarkers. For instance, the study of the cerebrospinal fluid (CSF) has added important advantages in understanding the pathophysiology of brain disorders [10]. However, considering that the lumbar puncture is an invasive method and that the composition of the CSF does not exactly reflect the neurochemistry in brain cells, this procedure is rarely used in psychiatric disorders, with most data related to affective or psychotic disorders [11] and limited evidence about ADs.

Blood sampling is potentially one of the most promising peripheral biomarkers because it is easily accessible and several neurotransmitters, neuropeptides, and neurotrophic factors are transported across the blood–brain barrier and reach the peripheral circulation, potentially reflecting the biological mechanisms of the CNS. However, it is not always possible to draw definitive inferences from the neurochemical composition of the blood to the activity of brain cells [3]. Moreover, different systems are involved in the pathogenesis of ADs, like the hypothalamic–pituitary–adrenal (HPA) axis [12] and the immune system [13], and the potential alterations underpinning these systems can be detected through peripheral sampling (e.g., through blood, saliva, urine, or hair).

Considering this background, a better knowledge of peripheral biomarkers seems crucial for clinicians. Therefore, the present review article will discuss the literature findings on peripheral biomarkers of the most representative DSM-5 (Diagnostic and Statistical Manual of Mental Disorders, 5^th^ edition) ADs—general anxiety disorders (GAD), panic disorders with or without agoraphobia (PDA), and social anxiety disorder (SAD)—with the aim to provide an updated overview of the topic. 

For the purpose of the present study, a literature search was conducted on the PubMed database considering articles published up until April 2020. Key search queries included combinations of the following terms: “biomarkers”, “anxiety disorders”, “peripheral biomarker”, “blood biomarker”, “generalized anxiety disorder”, “panic disorder”, “social anxiety disorder”, and “psychiatry”. In addition, reference lists of the selected articles were screened for additional research. We only included those articles published in English and that focused on the most investigated peripheral biomarkers related to neurotransmitters, neuropeptides, the HPA axis, neurotrophic factors, and the inflammation and immune system.

## 2. Neurotransmitters

Serotonin (5-HT), norepinephrine (NE), epinephrine (E), dopamine (DA), and gamma-aminobutyric acid (GABA) are the most investigated neurotransmitters in ADs (Figure 1). 

The monoaminergic systems (NE, E, DA, and 5-HT) have long been suggested to play a major role in the pathogenesis of ADs [14]. NE is an important neurotransmitter involved in the autonomic nervous response that is directly responsible for anxiety symptoms, usually associated with an increased NE metabolism and function [3,15]. 5-HT plays a determinant role in ADs, with two principal sources and systems involved: the amygdala-mediated threats, presumably linked to the emotion named “anxiety”, characteristic of GAD and obsessive-compulsive disorder (OCD) [16,17]; and the periaqueductal-grey-mediated threats, related to the emotion named “fear”, more closely related to the phobic, escape-dominant behavioral syndromes, such as specific phobias, SAD, and PDA [16,17]. Most of the 5-HT is produced outside the CNS and various peripheral organs express 5-HT receptors. 5-HT-receptors in blood cells (i.e., platelets, lymphocytes) are easily accessible and manifest certain similarities with CNS mechanisms that can be monitored through the measuring of peripheral parameters [18].

GABA is the most important inhibitory neurotransmitter system [19] and preclinical and clinical studies supported the link between the pathogenesis of ADs and dysfunctions of the central inhibitory mechanisms [20]. This evidence is also supported by the effective clinical response to BDZs, which act on the GABA system. Translocator protein (TSPO), a protein mainly found on the outer mitochondrial membrane, is considered a peripheral binding site for BDZ [21] and has a role in steroid biosynthesis, including neuroactive steroids, which exert anxiolytic properties [22]. Therefore, this molecule has been investigated as a treatment biomarker of ADs [3].

The next paragraphs collect the literature findings related to the neurotransmitters involved in ADs (GAD, PDA, and SAD) and Table 1 summarizes the investigated biomarkers and related results.

### 2.1. General Anxiety Disorders

The Serotonergic System: A study reported that platelet the 5-HT reuptake binding density was decreased in patients with GAD compared to healthy controls (HCs), even though this difference was reported also in patients with major depressive disorder (MDD), dysthymia, and PDA, therefore not specific to GAD [23]. Controversially, in another investigation, no differences in platelet 5-HT reuptake binding site density or affinity emerged among GAD, PDA, and HCs [24]. Another study assessing the 5-HT transporter binding density and affinity expressed on lymphocytes reported no difference between GAD patients and HCs [25]. Moreover, the same study investigated both the 5-HT and 5-hydroxyindoleacetic acid (5-HIAA, a metabolite of 5-HT) concentration in platelet-rich and -poor plasma and in lymphocytes, reporting no difference between GAD and HCs [25].

The effects of neurotransmitter modulations have been investigated as well. An acute tryptophan depletion technique that transiently lowers brain 5-HT was examined in clinically remitted patients with ADs [62]. Patients with GAD did not show psychological and physiological exacerbation of anxiety symptoms in response to stressors while patients with PDA, SAD, and post-traumatic stress disorder (PTSD) did [62].

The Noradrenergic System: Reduced platelet alpha-2 adrenergic peripheral receptor binding sites have been reported in GAD compared to the HCs [26,27]. Some studies investigated the effect of stimulating the noradrenergic system in patients with ADs. Abnormal changes in measures of anxiety, somatic symptoms, blood pressure, plasma NE metabolites, and cortisol levels emerged in patients with PDA but not in patients with GAD [14,63]. These results indicate different abnormalities in the regulation of the 5-HT and NE systems in GAD patients compared to other ADs, and these differences might be related to the abovementioned functional difference between anxiety and fear disorders [3].

The GABAergic System: A lower number of peripheral BDZ binding sites on platelets [28] and lymphocytes [29,30] has been reported in subjects with GAD compared to the HCs. Moreover, in two investigations of GAD patients compared to the HCs, treatment with BDZs showed to increase the number of peripheral binding sites, which also corresponded to improvement of anxiety symptoms [28,30]. 

### 2.2. Panic Disorders with or without Agoraphobia

The Serotonergic System: 5-HT plasma levels were reported to be significantly lower in PDA patients compared with HCs in some studies [24,31,32], while platelet 5-HT concentrations were not significantly different between PDA and HCs in other reports [33,34]. Moreover, platelet aggregation in response to 5-HT was significantly lower in PDA patients compared with the controls [35]. Platelet 5-HT uptake was measured in different studies, with inconsistent results. Indeed, four studies reported its decrease in PDA patients compared to the HCs [23,35,36,37], some studies reported no difference [24,38,39,40,41,42,43], while others reported an increase [42,44].

In CSF, 5-HIAA emerged to be significantly decreased in patients with a positive response to tricyclics, even though no difference at baseline was reported compared to the HCs [45]. On the other hand, in female patients with major depressive disorder (MDD) and comorbid PDA, a significant increase in CSF 5-HIAA levels was found compared with patients without comorbid PDA and with HCs [46]. Lastly, a significant increase in brain 5-HT turnover, computed via the jugular venous overflow of 5-HIAA, was observed in non-medicated PDA patients compared with HCs [47]. If assessed through a less invasive procedure (e.g., a blood sample), 5-HIAA might be a potential diagnostic biomarker of ADs (considering its increase compared to the HCs [46,47]) and, additionally, a predictive biomarker for treatment response [45].

Coplan and colleagues measured the antibodies directed at the 5-HT system, finding significantly elevated levels of plasma anti-serotonin and serotonin anti-idiotypic antibodies (directed at 5-HT receptors) in PDA patients compared with HCs, suggesting an autoimmune mechanism targeting the 5-HT system in this disorder [48].

The Dopaminergic System: Several studies using dopamine agonists suggested that this monoamine might be involved in the pathogenesis of ADs. In an epidemiological study, the risk of panic attacks was greater in cocaine users vs. non-users [64] and a significantly higher level of growth hormone (GH) in response to apomorphine (a dopaminergic agonist) emerged in PDA compared to depressed patients [50]. Previously, a report found that CSF levels of homovanillic acid (HVA), a metabolite of dopamine, were similar in PDA patients compared with HCs [45]. In a more recent study, the basal plasma dopamine levels were significantly higher in PDA patients compared with HCs [49].

The Noradrenergic System: Specific abnormalities in the regulation of the NE system in patients with PDA have been suggested [14]. The stimulation of the noradrenergic system produced abnormal changes in anxious symptoms and plasma NE metabolites in PDA patients but not in subjects with GAD, OCD, depression, or schizophrenia [63]. Elevated platelet aggregation in response to NE, platelet α-2-receptor density, and lymphocyte β-receptor density were found in patients with PDA treated with tricyclics [35], and these peripheral biomarkers have been proposed as potential trait markers in PDA. Cameron and colleagues [51] studied the platelet α-2-adrenoceptors using clonidine and yohimbine binding assays and their correlation to symptom severity and to NE and E plasma levels in the lying and standing position. Tritiated clonidine binding was decreased in PDA patients before pharmacological treatment, compared to the HCs, and the magnitude of binding decrease was correlated with symptom ratings and standing plasma NE. Moreover, PDA patients showed significantly increased standing plasma NE compared to the controls. Therefore, abnormal reactivity of NE to standing might be assumed as a biomarker of an increased likelihood of panic development [51]. In the same way, another study showed an increased α-2-adrenoceptor density in both lying and standing PDA patients [52].

Moreover, a low pre-treatment β-adrenoceptor affinity [65] and a decrease in plasma 3-methoxy-4-hydroxyphenylglycol (MHPG), the main central NE metabolite [66], have been reported as potential biological predictors of treatment response in patients with PDA. However, these results could not be confirmed in a study of the effects of imipramine in PDA, where the MHPG levels initially fell after starting the medication but returned to the pre-treatment levels by Week 8 of treatment [67]. The MHPG levels, measured in the CSF, were shown to be elevated in subjects with alcohol-use disorder and comorbid PDA [53], but a study on the effects of imipramine in PDA and another one conducted on PDA patients vs. HCs did not confirm this result [45].

Finally, plasma concentrations of E and DA, but not NE, were significantly higher at baseline in PDA patients compared to the HCs and, after treatment with paroxetine, the plasma catecholamine (E, NE, DA) levels showed a tendency toward a decrease. These results suggested a possible baseline increase in the plasma catecholamine levels in patients with PDA, which may normalize after appropriate treatment [49], as suggested by the selective serotonin reuptake inhibitors’ (SSRIs) effect on catecholamines, peripherally and centrally [68].

The GABAergic System: Nutt and colleagues suggested an underlying alteration in BDZ receptor sensitivity, reporting that subjects with PDA, compared to the controls, manifested more severe panic attacks after flumazenil intravenous injection [54]. However, other authors did not confirm these results [69]. Moreover, a dysfunction of GABA-A receptor modulatory neuroactive steroid regulation was observed by Rupprecht and colleagues [70]. Neuroactive steroids are derivatives of progesterone that modulate neuronal excitability through rapid nongenomic effects at the cell surface, acting on the GABA receptors [71]. PDA patients showed increased concentrations of GABA agonistic 3α-reduced neuroactive steroids [55], which may represent a counter-regulatory mechanism against the occurrence of spontaneous panic attacks. Differently, during experimentally induced panic attacks, PDA patients, compared with HCs, showed a significant decrease in GABA agonist 3α-reduced neurosteroids in association with an increased antagonistic 3α-reduced isomer [56].

In an experimental study, a specific TSPO ligand that enhances GABA-mediated neurotransmission was found to improve panic symptoms without causing sedation or withdrawal symptoms [72], hinting at TSPO ligands being possible candidates for new anxiolytic drugs. Moreover, peripheral BDZ receptors expressed on platelet membranes emerged to be significantly lower in patients with PDA than in HCs or subjects with OCD, suggesting that this peripheral biomarker may help differentiate some subtypes of these disorders [57].

### 2.3. Social Anxiety Disorder

The Serotonergic System: Platelet 5-HT2 receptor density did not differentiate SAD patients from the controls but was associated with panic attack severity [58]. Following a challenge with single doses of the partial serotonin agonist oral meta-chlorophenylpiperazine (mCPP), SAD patients did not significantly differ from HCs or OCD individuals in prolactin response to mCPP [59].

Tancer and colleagues evaluated the neuroendocrine correlations after challenges with the serotonergic (fenfluramine), dopaminergic (levodopa), and noradrenergic (clonidine) action molecules in SAD patients. An increased cortisol response to acute fenfluramine administration was observed in SAD patients but not in HCs. No other differences in the stimulus response emerged in SAD patients: no prolactin response to fenfluramine, nor a growth hormone or NE response to clonidine, nor prolactin or eye-blink responses to levodopa [60]. 

The Dopaminergic System: After challenging the dopaminergic system with the antagonist sulpiride and the agonist pramipexole, patients with SAD showed heightened anxiety symptoms but, after SSRI treatment (with remission of anxiety symptoms), patients showed an attenuated impact of pramipexole, suggesting a degree of dopamine D3 receptor desensitization after SSRI therapy [61].

## 3. Neuropeptides

Neuropeptides are defined as small protein-like molecules produced and released by neurons through the regulated secretory route and acting on the neural substrates [73,74]. Neuropeptides’ role in ADs have been extensively studied in animal and human samples, underlying their role in the pathophysiology of ADs and as promising new targets for treatment interventions [75,76]. The most important neuropeptides that play a role in the modulation of stress-related behaviors and anxiety are cholecystokinin (CCK) [77,78,79], oxytocin (OXT) [80,81,82,83], substance P [84], neuropeptide Y, galanin [76], pituitary adenylate activator polypeptide (PACAP) [85,86], ghrelin [87], and leptin [88]. 

CCK is one of most abundant neuropeptides in the brain and the CCK-B receptors are found with high densities in the hypothalamus, limbic system, basal ganglia, hippocampus, cortex, and brainstem. Several studies investigated the direct role of CCK in modulating anxiety and stress responses in human subjects [77,89].

OXT is a neuropeptide synthesized by the hypothalamus that regulates the activity of numerous brain structures (i.e., amygdala, hippocampus, striatum, cingulate cortex, and ventrolateral septum) [90]. Additionally, OXT has different important peripheral roles (e.g., muscle contraction during birth and milk ejection [91]). OXT central activity, mediated through the amygdala and the anterior cingulate, plays a central role in human social behavior, social cognition, anxiety, mood regulation, stress modulation, and fear learning and extinction [92]. The role of OXT has been investigated in different ADs [93], and some medications targeting the OXT system have been tested [94].

Ghrelin is a neuropeptide primarily involved in food intake that additionally influences emotions, mood, and anxiety regulation [87]. In several studies, ghrelin has been reported to induce anxious effects [95] and increased ghrelin secretion in stressful conditions determines anxious behaviors and activation of the HPA axis [96]. Leptin, an adipose-derived peptide hormone, mainly regulates energy balance and also modulates several CNS functions, comprising learning, memory, and mood and anxiety regulation [88].

The concentration of these neuropeptides can be monitored in the CSF and plasma samples, suggesting potential roles as peripheral biomarkers. The next section collects the literature findings related to the neuropeptides implicated in ADs (GAD, PDA, and SAD) and Table 2 summarizes the investigated biomarkers and associated results.

### 3.1. General Anxiety Disorders

Cholecystokinin (CCK): A study by Brawman-Mintzer [97] reported that intravenous pentagastrin, a CCK-B receptor agonist, induced higher rates of panic attacks in patients with GAD compared with the controls, suggesting that CCK hypersensitivity could play a role in the pathophysiology of this disorder. Different preclinical studies showed an anxiolytic effect of CCK-B antagonists [108,109,110,111,112]: clinical studies in GAD patients, however, showed controversial results [113,114,115].

Pituitary Adenylate Cyclase-Activating Polypeptide. A study reported female GAD patients, but not male subjects, to be associated with lower levels of circulating PACAP compared with HCs, as well as with worse anxiety somatic symptoms and insomnia when associated with a specific gene polymorphism (PAC1R CC) [86]. This gender difference could be related to an estrogen response element of the PAC1R allele [85]. Other studies reported that altered PACAP peripheral levels may induce somatic symptoms of anxiety, including headaches and altered breathing [116,117].

Ghrelin: A recent study on ghrelin and leptin levels in children with ADs, including GAD, showed higher ghrelin levels in female patients compared to the HCs, but not in boys [98]. Moreover, a significant positive correlation between trait anxiety in girls and plasma ghrelin levels was found, providing indirect support to the gender difference in state and traits. This result suggests that ghrelin may play a role in the etiological mechanisms of ADs. However, more studies are needed to explain its role in GAD.

Leptin: Mixed results on the anxiolytic effects of leptin were reported. Ozmen and colleagues did not find differences in the circulating levels between children with GAD, SAD, and separation anxiety and the HCs [98]. However, many other studies have demonstrated higher leptin levels in patients with anxiety and psychological stress [118].

Oxytocin (OXT): The oxytocinergic system in GAD patients has not been extensively studied [80] and only a few studies have investigated the anxiolytic properties of intranasal OXT in GAD patients [93,94]. Bailey and colleagues [119] reported a significant reduction in anxiety symptoms after intranasal OXT administration following inhalation of carbon dioxide by 7.5% (proposed as a model of GAD), in a similar way to the BDZ lorazepam [120].

### 3.2. Panic Disorders with or without Agoraphobia

Cholecystokinin (CCK): Several studies emphasized CCK’s panicogenic effect in humans and this neuropeptide satisfied most of the criteria for an “ideal” challenge substance for PDA [121,122,123]. Cholecystokinin-4 (CCK-4), a CCK receptor agonist, has been extensively used to cause panic symptoms in patients and healthy volunteers, with some differences between groups. Indeed, the dose-response to intravenous CCK-4 was shown to reliably differentiate PDA patients from HCs [78]. Specifically, after injection of 25 μg of CCK-4, the panic rate was 91% for patients and 17% for HCs, and 50 μg induced a full-blown panic attack in 100% of patients vs. 47% of the controls. Moreover, CCK-panic induction might serve as a tool to assess the anti-panic potential of the anxiolytic compounds. Indeed, in CCK-4-sensitive healthy volunteers, a significant reduction was observed after administration of the BDZ alprazolam and the GABAergic anticonvulsant vigabatrin [124,125]. These results suggest that BDZ might exert their clinically relevant action through a mechanism involving antagonism of CCK-induced excitation. With respect to SSRIs, treatment with fluvoxamine significantly decreased the sensitivity of the PDA patients to CCK-4, compared to patients treated with a pill placebo. Moreover, in the responders’ group, 83% no longer experienced a CCK-4 induced panic attack, while in the non-responders’ group this was only 28% [126]. Another investigation on healthy subjects showed a significant reduction in CCK-4-induced panic rates without significant differences between the group treated with escitalopram and the placebo condition [127]. This discrepancy might be explained considering the possible role of the placebo effect and habituation on the pharmacological modulation of CCK-4-induced panic and, therefore, to be cautious in applying a CCK-4 challenge to screen the anti-panic properties of new drugs [127].

Additionally, lower concentrations of Cholecystokinin-8 (CCK-8) in CSF [99] and in peripheral lymphocytes [100] have been reported in PDA patients but not in healthy subjects, showing abnormalities in the entire CCK system (CCK-4 and CCK-8) and underpinning the pathophysiology of anxiety and panic disorder.

Atrial Natriuretic Peptide (ANP): A role for ANP in ADs has been hypothesized, due to its role in inhibiting the corticotropin-releasing hormone (CRH)-stimulated release of the ACTH hormone [128] and cortisol [129], overall reducing the stress-response activation of the HPA system [130]. Moreover, compared to the HCs, lower basal ANP plasma levels characterize PDA patients, but at the same time the ANP release was more pronounced during an experimentally provoked panic attack [103]. In patients with PDA, ANP infusions (compared to placebo) decreased the CCK-4-induced panic attacks [131] and a significantly accelerated release of ANP has been described in patients with lactate-induced panic attacks compared to the HCs [104].

Oxytocin (OXT): It has been emphasized that the OXT role in PDA lies in increased rationalization and threat processing in a top-down manner, influencing dysregulated neuronal panic networks [132]. However, no investigation directly analyzed the peripheral OXT levels in PDA patients [80].

Leptin: Leptin is a peptide hormone mainly produced by white adipose tissue [133], which is localized in several brain structures [134], suggesting a possible role in modulating neurobiological and psychological processes [135,136]. Leptin could be protective in the development of PDA through its modulatory action on the HPA axis, promoting a reduction in the release of corticosterone regulated by ACTH [101]. Similar baseline plasma levels of leptin have been reported in PDA patients vs. HCs [101]. However, limited to female patients, lower leptin serum levels were associated with a greater severity of psychopathological manifestations, including the number of panic attacks, symptoms of somatization, anxiety and phobic anxiety, and overall clinical presentation compared to the HCs [102]. Moreover, in patients with MDD treated with different antidepressants, higher pretreatment leptin was associated with a better response in treating panic symptoms [101].

Adiponectin: In addition to adiponectin’s principal role in regulating body metabolism, this neuropeptide has shown a potential involvement in the pathogenesis of some psychiatric disorders, such as anxiety and panic disorder. This hypothesis was investigated in a small study, showing PDA patients with significantly lower plasma levels of adiponectin compared to HCs [105]. However, this result was not confirmed in a more recent study [101], reporting no differences in terms of adiponectin plasma levels between PDA patients and HCs.

### 3.3. Social Anxiety Disorder

Oxytocin: A direct comparison of OXT plasma levels between SAD patients and HCs did not yield significant results [106]. However, in another investigation, a higher severity of social anxiety symptoms positively correlated with higher OXT plasma levels in SAD patients but not among HCs [107]. This might be because higher OXT secretion is an insufficient compensatory attempt to reduce social anxiety symptoms. Adjunctive OXT to exposure therapy in patients with SAD showed improved mental representations of the self and better evaluations of speech performance compared to the HCs, though without improvement in overall treatment outcomes from exposure therapy [132].

## 4. Hypothalamic–Pituitary–Adrenal Axis

The HPA axis is the major neuroendocrine mediator of the stress response. Figure 2 shows a brief description of the HPA axis regulation.

The combination of genetic factors, early-life stressors, and repetitive trauma has a role in the dysregulation of this system that may subsequently result in increasing individual vulnerability to ADs [137,138]. The HPA axis’ functions and modifications can be easily investigated thanks to numerous peripheral biomarkers using different samples (i.e., plasma, saliva, urine, and hair) of the involved hormones and neurotransmitters (i.e., cortisol, NE and E), releasing factors, and catabolites. However, the large heterogeneity in measuring the cortisol stress response and the complexity of this system has led to different, and sometimes inconsistent, results. Trying to overcome these obstacles, a systematic review analyzed the different cortisol response measures to psychosocial stress in ADs [139], using standardized cortisol outcomes (the areas under the curve with respect to increase (AUCi) and ground (AUCg)) as a biomarker. As a result, a gender- and phase-specific cortisol reactivity emerged, with women showing a “blunted” cortisol stress response compared with the HCs and men with SAD showing an elevated cortisol response. These results indicate the important influence of different variables on the cortisol response and how different specific ADs might have a specific impact. 

The next paragraphs include the literature findings related to the HPA axis in ADs (GAD, PDA, and SAD) and Table 3 summarizes the investigated biomarkers and related results.

### 4.1. General Anxiety Disorders

Summarizing the results from different investigations, it remains uncertain whether GAD is associated with abnormally increased cortisol levels. In this respect, in a study collecting male adolescent GAD patients, similar cortisol plasma levels at the baseline and after a stressful test was reported in GAD patients and the HCs. In the same report, the pre-stress ACTH concentration was higher in patients compared to the HCs, with similar levels after the test [140]. In a large study of 1427 GAD patients and HCs, the former displayed a significantly greater cortisol awakening response, but only in the case of comorbid MDD [148]. Among Vietnam veterans, subjects suffering from GAD showed cortisol and dehydroepiandrosterone (DHEAS) plasma levels and a cortisol/DHEAS ratio similar to the HCs [141]. Additionally, a sample of children with GAD did not differ from the HCs in relation to pre-sleep salivary cortisol levels, despite the presence of altered sleep patterns [202].

Salivary-baseline cortisol levels of elderly subjects with at least one ADs (including GAD) were comparable with those of the HCs. However, when exposed to common stressful situations, male subjects showed a slower decline rate of post-stress cortisol increases compared with the controls, while in females the clinical severity was associated with a larger post-stress cortisol secretion and lower recovery capacity following the stressful situation [143]. According to the authors, women’s increased reactivity to stressful environmental conditions might be explained by their overall heightened vulnerability to anxiety disorders.

Differently, higher plasma [203] and salivary cortisol levels were overall significantly more elevated in GAD patients than the HCs and were positively associated with the GAD symptoms [144]. On the other hand, some studies showed a reduction in the HPA activity. The hair cortisol concentration, which reflects the long-term cortisol levels independently of the acute HPA axis responses, has been reported to be up to 50–60% lower compared with the HCs [145,146]. These results suggest that chronic anxiety may result in sustained and prolonged downregulation of the HPA axis activity. Indeed, a study reported that adults over 65 years of age with long-lasting ADs displayed a lower cortisol awakening response compared with HCs and this correlation was most prominent in GAD patients [149].

Additionally, the ratio of salivary alpha-amylase (sAA, a biomarker of chronic stress linked to the sympathetic nervous system) over salivary cortisol has been investigated. Compared to the HCs, GAD patients exhibited a greater baseline ratio of sAA/cortisol and a smaller ratio of sAA/cortisol following a mental arithmetic challenge. These results lead to hypothesize an asymmetry between the autonomic nervous system and the HPA axis in GAD, suggesting that increased sympathetic nervous system suppression in GAD may be partially mediated by cortisol activity [147].

Lastly, different studies investigated the HPA axis modifications (suppression and response to treatment) as possible biomarkers. In this light, non-suppression in the dexamethasone suppression test emerged to be comparable to what was observed in MDD outpatients, though with limited value in differentiating between GAD and other ADs [142,157,158,159,160]. Cognitive-behavioral therapy (CBT) [150], escitalopram [151], and treatments refocusing GAD patients’ attention [152] were found to be followed by reduced cortisol levels compared to the HCs. Controversially, after treatment with buspirone [153], alprazolam [154], or diazepam [155,156], no correlations between treatment response and post-treatment changes in cortisol levels, or no change in the cortisol levels at all, were reported.

### 4.2. Panic Disorders with or without Agoraphobia

It was suggested that panic attacks might be a consequence of the disruption of the HPA axis, caused by a dysfunctional response to stressful events [204]. Several studies investigated the basal differences in blood cortisol concentrations with inconsistent results. Indeed, compared with the HCs, PDA patients showed higher cortisol concentrations during the day [161,162,163] or the night [164], while other reports described a similar concentration between patients and the controls [165,166]. Urinary free cortisol levels appeared to be normal [167], elevated [168], or increased only in patients with complicated PDA [169] compared to the HCs.

Contradictory data about the dysregulation of the HPA axis emerged in some studies. The plasma ACTH concentration was increased in patients compared with the controls at the baseline [166]. After the HPA axis stimulation tests, a lower ACTH response to corticotropin-releasing hormone (CRH) was reported in patients compared with the HCs in three studies [162,166,171] and normal responses in one [172]. The cortisol values after CRH were found to be reduced in two [162,166] and normal in two other reports [171,172].

Additionally, HPA axis modification during panic attacks showed inconsistent findings. After spontaneous panic attacks, non-significantly elevated plasma cortisol levels emerged in one study [173], but significantly increased salivary cortisol was reported in another investigation [185]. After exposure to feared situations, PDA patients did not show increased plasma cortisol and ACTH concentrations [174]. During panic attacks induced by lactate infusion, most studies did not show elevations in ACTH or cortisol [175,176,177,178,179,180]. Differently, one study showed marginally higher cortisol levels than the controls [181] and another investigation reported decreased cortisol levels during lactate-induced panic attacks in the patients and controls [59].

Compared to the HCs, PDA patients showed a significantly larger increase in plasma cortisol after yohimbine-induced panic attacks [182]. By contrast, mCPP or caffeine did not differentiate between the PDA and HCs, on the basis of similar plasma cortisol levels observed in both groups after the induced panic attack [183,184].

Considering the HPA axis modifications as biomarkers of treatment response in PDA, a reduction of salivary cortisol levels has been reported after exercise training [205]. On the contrary, nocturnal urinary cortisol excretion did not differ in response to paroxetine or a placebo combined with relaxation training or aerobic exercise [170].

### 4.3. Social Anxiety Disorder

Similar baseline cortisol levels or cortisol responses after pharmacological or psychological challenges were reported in SAD patients compared with the HCs using different biomarkers: free cortisol levels [186,188]; the free cortisol/creatinine ratio [186]; 24-h excretion of urinary free cortisol and post-dexamethasone cortisol levels [187]; diurnal saliva cortisol levels [188,191]; cortisol increases observed before social stressing situations (in female adolescents) [191]; and scalp-near hair samples [188]. Moreover, significantly greater ACTH and cortisol responses to stress [200] and a significantly greater cortisol awakening response [148] emerged in SAD patients compared to HCs only when comorbid with MDD. The levels of cortisol were not significantly different in the patients compared to the controls after the intravenous administration of citalopram, CCK-4, or mCPP [194,195,196].

On the contrary, the different baseline and/or after cortisol responses challenge the distinguished SAD patients from the HCs in some studies [197]. Administration of fenfluramine [206] or mCPP [59] or exposure to an arithmetic/working memory task in front of an audience [198] or to a speech-stressor [199] resulted in significantly greater cortisol responses compared with the HCs. For children of 4.5 years of age, an elevated afternoon salivary cortisol level was reported as a risk factor for chronic high inhibition in school age and SAD occurrence in adolescence [192].

Additionally, in adolescents followed up to 6 years, a higher baseline cortisol awakening response was reported as a strong and significant predictor of the subset of SAD onsets [193]. Lastly, a pattern of elevated sympathetic activity, reduced parasympathetic, and reduced HPA axis activity was reported among children with an AD (including SAD) compared with the controls, resembling the traits of a chronic stress condition [207].

Different studies showed increased levels of salivary α-amylase and this finding led some authors to suggest that the SAD psychopathology might be related to a vulnerability of the autonomic nervous system, rather that of the HPA axis [189,190,201]. However, Kramer and colleagues, in a study with SAD children, found that the salivary cortisol levels showed significantly higher reactivity compared with α-amylase after undergoing a specific stress test [208].

## 5. Neurotrophic Factors

Different neurotrophins manifested specific roles in the pathogenesis of ADs, in particular the brain-derived neurotrophic factor (BDNF) [209,210], nerve growth factor (NGF) [211,212], fibroblast growth factor-2 (FGF2), glial cell-line-derived neurotrophic factor (GDNF) [213], neurotrophin-3, neurotrophin-4, and artemin [3].

NGF is a neuropeptide involved in regulating the neuron growth, maintenance, proliferation, and survival of certain target neurons [214]. In a sample of healthy subjects, an association between trait anxiety and a genetic variation of NGF was reported [215]. Interestingly, NGF was increased during and after the jump in soldiers making their first parachute jump [216]. A reduction in NGF has been consistently reported in depressed subjects [217], but its role in patients with ADs has not been widely investigated [3].

BDNF is a neurotrophin involved in the synaptic plasticity and survival of neurons in the brain and in the peripheral nervous system. It has been assumed that BDNF is implicated in the etiology of depression and ADs. However, findings of BDNF protein levels in ADs remains inconsistent [218].

FGF-2, a protein involved in neuroregeneration and stress regulation, has been recently proposed as a promising biomarker for anxiety and trauma disorders. Indeed, a lower serum and salivary FGF-2 was correlated with greater fear responses to both threatening and safe stimuli in healthy participants undergoing a differential fear conditioning procedure, being considered as a potential biomarker for AD vulnerability [219].

The next section reports the literature findings related to the neurotrophic factors involved in ADs (GAD, PDA, and SAD) and Table 4 comprises the investigated biomarkers and associated results.

### 5.1. General Anxiety Disorders

BDNF: Many studies suggested a BDNF involvement in the neurobiology of GAD and its role as a biomarker [210,211,212,213,214,215,216,217,218,219,220,230]. However, unlike the consistent results in major depressive disorder, findings related to BDNF in GAD remain controversial. In a large sample of patients with different ADs, including GAD, no changes in the BDNF levels were found when compared with the HCs and regardless of type of AD [221]. No significant association between the baseline plasma BDNF levels and the severity of the disorder emerged in another investigation with GAD patients [222]. An Italian study showed similar serum BDNF levels in GAD patients compared to HCs, but a significantly lower BDNF level if only females were analyzed [220]. This gender influence is partially in agreement with the proposed hypothesis that serum BDNF is altered only in females with multiple types of ADs [231]; however, the reason for the gender-specific association of reduced BDNF levels in female GAD patients remains presently unknown and warrants future investigations [220].

By contrast, significantly lower BDNF plasma levels in GAD patients vs. the HCs emerged in other reports [210,223,224]. Shen and colleagues observed higher baseline plasma BDNF levels in GAD patients vs. controls and a normalization after GAD remission (patients were treated with paroxetine) [225]. Additionally, higher BDNF levels were found in the cord blood of newborn infants of women with GAD, compared to healthy women [226]. Lastly, when comorbid with MDD, GAD patients showed doubled plasma levels of BDNF and artemin, a glial cell-line-derived neurotrophic factor family member, compared to the HCs [230].

Some studies combined BDNF and glial cell-derived neurotrophic factor (GDNF) to explore their relationship with GAD, investigating the characteristics of serum and their potentials to predict treatment remission in GAD patients [224,228]. Indeed, serum BDNF/GDNF levels were lower in GAD patients compared to HCs and, specifically, females showed higher BDNF/GDNF levels compared to males [224]. According to these authors, the gender-related influence might be related to the influence of menstrual cycle, considering that most of the subjects included were female in the menopausal period. 

In a meta-analysis, a GDNF downregulation in GAD patients was reported, although the difference compared to the HCs was not as significant, as observed for the BDNF level [228].

NGF: A study reported that the NGF serum levels of the patients and controls were similar at the baseline. However, after successful CBT treatment (the Hamilton Anxiety Scale score changed from 22.23 ± 4.19 before treatment to 9.41 ± 6.87 (*p* < 0.0001) after treatment), the patients’ NGF serum concentrations increased significantly, which might correspond to an altered stress reaction, possibly contributing to a positive therapeutic response with CBT [227].

### 5.2. Panic Disorders with or without Agoraphobia

BDNF: Several studies investigated the role of BDNF in the etiology of PDA [229,232,233]. No differences were found in the BDNF plasma levels compared to the HCs and to patients with MDD [101]. In some reports, a lower serum BDNF level was associated with a greater risk to develop panic symptoms, and with an inadequate treatment response [218,229], but other authors did not support BDNF as a biomarker for panic attacks [101,234,235]. Moreover, serum BDNF levels were hypothesized to be predictive of treatment success in PDA patients. BDNF plasma levels were lower in PDA patients with a poor response to CBT compared with those who showed a better response [229]. Furthermore, after 30 min of aerobic exercise, significantly higher blood BDNF levels were observed in PDA patients compared to the HCs [232].

NGF: Salles and colleagues [223] investigated the possible correlation between NGF and mental disorders, showing no significant differences in serum NGF levels in PDA patients [223].

GDNF: Pedrotti Moreira and colleagues tested the potential association of GDNF with different ADs, finding significant higher serum concentrations of GDNF in PDA patients compared to the HCs [213].

### 5.3. Social Anxiety Disorder

BDNF: In a large sample of patients with different ADs, including SAD, Molendijk and colleagues showed lower levels of BDNF among female patients compared to the controls, but this finding was not replicated in the general sample and, thus, peripheral BDNF concentrations did not have enough specificity to categorize a specific AD [221].

GDNF: In the aforementioned study, SAD patients were included as well and significantly higher serum GDNF values, compared to the HCs, were reported [213].

## 6. Inflammation and immune system

The role of the immune system and inflammation processes in the pathophysiology of ADs has been reported in several studies [3,236,237,238], also supported by the high rate of comorbidity between ADs and several inflammatory medical conditions [239,240]. Different potential biomarkers of inflammation and the immune system have been investigated, including cytokines (interleukins (ILs), tumor necrosis factors (TNFs), and interferons (IFNs), cells (phagocytes, lymphocytes), and antibodies, suggesting their potential role as biomarkers in the diagnosis of ADs [3]. In a national database with 2861 participants, higher C reactive protein (CRP) levels were similarly associated with somatic and cognitive symptoms of anxiety in male subjects, while IL-6 and TNF-a levels were associated with somatic symptoms of anxiety. For all the reported associations, lifestyle played an important role, with the body mass index explaining most of the relationship, underling the presence of common pathophysiological mechanisms between inflammatory medical conditions and anxiety [241]. However, not all available reports hinted at a positive association between inflammation and anxiety symptoms, suggesting other factors may interfere with inflammation in the expression of these disorders (i.e., gender, comorbid conditions, types of trauma/stress exposure, and behavioral sources of inflammation) [237].

One explanation of the increased inflammation in ADs is their strong associations with the stress response mechanisms mediated by the HPA axis and the autonomic system. NE and E directly modulate the release of cytokines and inflammation through adrenoceptors on immune cells [242]. When stressful situations are prolonged, a dysregulation of the stress axis occur that determines an additional inflammation and contributes to increased symptoms by direct effects on specific brain regions (such as the prefrontal cortex, insula, amygdala, and hippocampus), deemed critical for the regulation of fear and anxiety [237].

Oxidative stress is another marker extensively investigated in psychiatric disorders, consisting of different mechanisms regulating the balance between reactive oxygen species (ROS) and antioxidant defenses [243]. Altered oxidative mechanisms may be responsible for the pathophysiology of psychiatric disorders as well and explain the overlap between ADs and other inflammatory-based diseases.

Taken as a whole, the literature data indicate a clear role for inflammation in the etiology and maintenance of ADs. Nonetheless, increased inflammation is not specific to these conditions and can be seen in other psychiatric disorders like depression [244].

The next paragraphs collect the literature findings related to the inflammation and immune systems involved in ADs (GAD, PDA, and SAD) and Table 5 shows the investigated biomarkers and related results.

### 6.1. General Anxiety Disorders

The correlation between anxiety and inflammatory medical conditions has been specifically investigated in GAD, with prospective epidemiological studies reporting an association with impaired immune function and increased risk for cardiovascular diseases or events [261]. Costello and colleagues [245] conducted a systematic review and meta-analysis of the peripheral inflammatory cytokines in GAD, including data about 16 different cytokines. Peripheral levels of CRP, INF-γ, and TNF-α were significantly raised in patients with GAD compared with the controls in two or more studies. Ten further proinflammatory cytokines (IL1, IL-1α, IL-2, IL-6, IL-8, IL-10, IL-12p70, monocyte chemoattractant protein-1, stromal derived factor-1, and granulocyte-macrophage colony-stimulating factor) were reported to be significantly increased in at least one study. Five out of 14 studies reported no difference in the levels of at least one cytokine. Considering the data from a meta-analysis, CRP was significantly higher in people with GAD compared with the controls, although with a small effect size, and comparable with that reported in schizophrenia but greater than in other ADs or MDD. These results shed light on the role of an inflammatory response in GAD, although the authors concluded that the role of inflammatory cytokines in GAD etiology remains unclear, with further longitudinal studies required.

With respect to oxidative balance, a higher total oxidant status and oxidative stress index [246,247] and a lowered total antioxidant status [246] were reported in GAD patients, compared to the HCs. Additionally, increased nitro-oxidative stress (i.e., nitric oxide (NO) production) has been reported [248].

Other studies found lowered levels of specific antioxidants or antioxidant enzymes, with inconsistent findings. Imbalanced peripheral superoxide dismutase and catalase have been reported, with increased activity in some studies [248] but decreased in others [249]. Paraoxonase 1 was found to be decreased in different studies [250] but not all [248,251]. Lowered free sulfhydryl groups, an important member of antioxidant defense mechanisms, was reduced in GAD patients compared to the controls [250]. Moreover, GAD exhibited altered lipid peroxidation, as reported by increased levels of lipid hydroperoxides [248,251], lowered lipid-associated antioxidant defenses, decreased HDL cholesterol [248], and increased malondialdehyde, the end product of lipid peroxidation [249]. Increased lipid peroxidation may interfere with the pathophysiology of GAD by causing a reduced membrane fluidity and damaged membrane proteins, which can alter neurotransmission, neuronal functions, and brain activities [262]. Lastly, increased uric acid levels have been reported in some studies as a compensatory mechanism against aldehyde production and oxidation [248].

### 6.2. Panic Disorders with or without Agoraphobia

Numerous studies reported altered cellular immunity in PDA. Two studies found evidence of alterations in circulating lymphocyte profiles and reduced cell activation [263,264]. Kim and colleagues [258] found that baseline peripheral lymphocyte subsets did not differ between PDA patients and HCs but, after paroxetine treatment (with a significant reduction in different psychometric scales), PDA patients showed a significant increase in the CD3+, CD4+, and CD8+ T lymphocyte proportions, and a decreased B lymphocyte proportion. These data suggest that the immunological variables are affected by pharmacotherapy in patients with panic disorder. Among these biomarkers, an increased percentage of CD8+ cells negatively correlated with the pretreatment Clinical Global Impression (CGI) score [258].

In a study by Schleifer and colleagues [259], drug-free PDA patients showed decreased percentages of CD19+ B lymphocytes, while natural killer cell activity did not differ with the HCs. Moreover, another investigation found that drug-free PDA patients did not differ from the HCs, except for significantly lower CD4+ cells [260]. In addition to immune cell counts, other indirect approaches related to cellular immunity have been tested. Yolac and colleagues [252] determined the serum levels of adenosine deaminase (ADA) and dipeptidyl peptidase IV (DPPIV), two enzymes involved in the activation of T lymphocytes and regulation of cellular immunity. ADA and DPPIV were significantly higher in the drug-free PD patients than in the HCs. Cell-mediated immune functions, measured by the lymphocyte proliferative response to phytohemagglutinin and IL-2 production, were found to be reduced in drug-free PDA patients compared with the HCs, but after treatment (combined psychotherapy and pharmacological approach) no significant differences were found in the two groups [265]. Moreover, in the treatment group, a reduction in the self-reported anxiety level after treatment was significantly associated with a change in the lymphocyte proliferative response to phytohemagglutinin, suggesting its role as a potential treatment response biomarker.

To overcome the various and inconsistent literature findings regarding cytokine’s role in PDA, Quagliato and colleagues [253] conducted a systematic analysis and summary of all the main published data on this issue. Serum levels of IL-6, IL-1β, and IL-5 were consistently reported to be altered in PDA patients compared with HCs, while findings about IL-2, IL-12, and INF-γ were variable. The authors concluded that the heterogeneity in these results may be attributable to variability in methodology and to differences between the studied populations [253]. Of note, after a psychosocial stress test, no differential secretion patterns of IL-6 emerged in PDA patients compared to the HCs, but the IL-6 peak reaction correlated significantly with disease severity and higher IL-10 levels distinguished PDA patients vs. the HCs [254]. 

Lastly, in a recent study, peripheral inflammatory cytokines have been related to the kynurenine (KYN) (breakdown product of tryptophan) pathway and to cognitive deficits [255]. In PDA patients, but not in HCs, IL-2 soluble receptor levels were significantly associated with serum KYN concentrations. Moreover, the KYN/tryptophan ratio emerged as a potential biomarker for PDA patients. Indeed, an elevated KYN/tryptophan ratio significantly predicted poor short-term verbal memory, suggesting that these subgroups of patients might have cognitive deficits, thus treatments targeting the kynurenine pathway may improve cognitive abnormalities in PDA patients [255].

Considering the humoral immunity, one study documented significantly lower levels of C3a, C5a, and C5b in PDA patients compared to the HCs, both before and after pharmacological treatment. However, with respect to complement molecules, no significant differences were found comparing the patient group before and after effective pharmacological treatment [256].

Additionally, in PDA patients, a large proportion (30%) of mannose-binding lectin (MBL)-deficient (<100 ng/mL) individuals was observed alongside a significantly lower level of MBL and MBL-associated serine protease-2 [257]. Since MBL deficiency is highly heterogeneous and associated with both infectious and autoimmune states, further research is needed to identify which complement pathway components are associated with PDA and which factors, alongside chronic stress, are responsible for a lowered concentration of these molecules.

### 6.3. Social Anxiety Disorder

In a general sample of patients with ADs, female SAD patients had lower plasma levels of CRP and IL-6 and the highest CRP levels were found in subjects with an older age at the AD onset [236]. Once again, the influence of sex-hormones has been implicated in this result, as sex differences become less clear with increasing age, as a result of hormonal changes across the women’s lifespan, which influence inflammation levels [236].

## 7. Conclusions

A first consideration relates to the observed change in paradigms considering that most of the studies investigating blood- or CSF-based biomarkers in ADs have been published in the 1980s and 1990s, while more recent studies shifted the research onto different potential biomarkers, mainly emphasizing neuroimaging or genetics. This shift might be explained, on one hand, by variable and not always consistent results in peripheral biomarkers studies and, on the other, by the development and introduction of diagnostic and sophisticated neuroimaging and genetic techniques. However, ADs are multigenic diseases and the contribution of single genes is only small [3,266]. Since many environmental risk factors are associated with ADs, the interactions between genes and the environment should be also considered; this, however, is out of the scope of the present review. The contribution of epigenetic mechanisms should be also taken into account as a translator of the environmental effects in several psychiatric conditions, including ADs, as already reported in some investigations [267,268].

For a correct interpretation of the results included in the present article, the impact of the different variables related to the study methodology (i.e., sampling, analytic procedures, and comorbid conditions with ADs) and limitations should have been considered and investigated; however, this was outside of the scope of the present overview, considering also the vast amount of selected articles. Moreover, in the present article the authors focused on the most investigated peripheral biomarkers in ADs; therefore, other molecules not herein mentioned might have been considered as biomarkers in other investigations.

Overall, despite promising results, most reports have produced solitary findings, sometimes inconsistent and not clearly replicable and applicable to clinical practice. In fact, even though these studies have provided a relevant contribution to increase the knowledge of the neurobiological mechanisms of ADs, no specific dysfunction of a particular neurotransmitter or a neuropeptide could be defined as the main cause for ADs and, consequently, be chosen as a diagnostic peripheral biomarker. Among other reasons, this might be related to the limited sample size of most studies in the field and to the influence of some clinical variables (i.e., gender, age, concomitant medication, comorbidities, and clinical severity) as well as to study methodology. Moreover, in some cases, the same biomarker showed to play a role also in other psychiatric disorders than ADs (e.g., BDNF in depression [269] or OCD [267], and OXT in depression or schizophrenia [270]), emphasizing how psychiatric disorders exist in a continuum expression of symptomatology and that the common genes and pathological mechanisms are underpinned.

In summary, biomarkers related to GAD showed some degree of consistency with respect to the GABAergic system, with most of the studies reporting a reduction in the GABA-related biomarkers in GAD patients [28,29,30]. Although the HPA axis is dysregulated also in GAD [138], no consistent results can be drawn with respect to the peripheral biomarkers, with some studies showing an increase, while others a decrease in the investigated biomarkers, at the baseline or after the stress test/treatments. BDNF showed some consistent results (reduction of basal levels in some reports [210,220,223,224]), likely supporting its role in different ADs and encouraging further investigation [218]. In PDA, several studies showed an increase in the peripheral biomarkers related to the noradrenergic system [35,49,51,52,53], underlining an abnormal noradrenergic system regulation with consequent pathological clinical expression. Among the neuropeptides, several findings are related to abnormalities in the CCK system in PDA patients [78,99,100], and potential new molecules (CCK-receptor antagonists) targeting this system have been investigated [109,112]. For SAD, findings related to peripheral biomarkers are mostly solitary, probably due to the relative minor epidemiologic and clinical impact of this disorder, therefore no peripheral biomarker of diagnostic value can be chosen.

Moreover, all peripheral biomarkers of treatment response remain in very early stages of development and only a few field studies have demonstrated reliability in predicting a pharmacological response. Results from the present article showed that some peripheral biomarkers related to neurotransmitters were associated with treatment response. Among these, the CSF 5-HIAA level in GAD patients emerged to be significantly decreased in patients with a positive response to tricyclics [45]. In addition, a low pre-treatment β-adrenoceptor affinity was associated with a positive response to paroxetine [65]. With respect to the HPA axis in GAD, a reduction in the cortisol levels was associated with a positive response to CBT [150], escitalopram [151], and treatments refocusing patients’ attention [152]. Some neurotropic factors showed a predictive value in treatment response. In GAD patients, a reduction in the BDNF levels followed a symptoms remission (after treatment with paroxetine) [224] and the NGF serum levels, after successful CBT, increased significantly [227]. Moreover, in PDA, a lower serum BDNF level was associated with an inadequate treatment response to CBT [229]. With respect to the inflammation and immune system, PDA patients showed a significant increase in CD3+, CD4+, and CD8+ T lymphocyte proportions, and a decreased B lymphocyte proportion after effective paroxetine treatment and, in the same disorder, a change in the lymphocyte proliferative response to phytohemagglutinin correlated with a reduction in self-reported anxiety level after treatment [265]. Although the abovementioned results mostly represent solitary findings, they constitute the first step for additional studies targeting these peripheral biomarkers.

Finding an appropriate biomarker in ADs, as well in other mental disorders such as depression, remains one of the most important unmet needs and goals for psychiatry research [271]. This issue was confirmed in the last edition of the DSM [272] where no biomarkers are reported (though their inclusion was a goal originally aspired to [273]) and remains being modelled on symptom clusters. The diagnosis of ADs depends nowadays solely on symptomatic and clinical information due to the lack of available objective biomarkers [274]. 

Thus, determining a practical biomarker in patient-derived peripheral samples that helps in the correct diagnosis, prognosis, and prediction of treatment response remains highly desirable. One of the first steps to increase the identification of a potential biomarker is to better define which “problem” is investigated. To do so, scientific searches should narrow their focus and clearly define what the biomarker is aimed to investigate (e.g., which disease phenotype, clinical diagnostic group, gender, or age). Following this purpose, possible biases related to the studies’ methodology could be overtaken. Consequently, multicentric investigational studies should be encouraged in order to collect a large number of patients with narrow identical baseline characteristics that accurately represent the larger population.

If the discovery of an ideal biomarker is achieved, its use in clinical practice should be further investigated and implemented, considering additional characteristics like costs, patients’ tolerability, and overall clinical applicability. To do so, further economic resources should be invested to implement the technologies that permit practical, affordable, and robust sampling and diagnostic techniques that can be routinely used in research and in clinical practice. 

Lastly, a future direction might consist of adopting a machine learning approach, which has been previously used to address mental health questions [275]. Peripheral biomarkers, combined with other biomarkers (related to genetic, epigenetic, neuroimaging, and neurophysiology) and clinical variables, might contribute to the machine learning algorithms that could calculate the risk to develop a specific AD, determine the clinical evolution, and find precise and personalized treatments.

## Figures and Tables

**Figure 1 brainsci-10-00564-f001:**
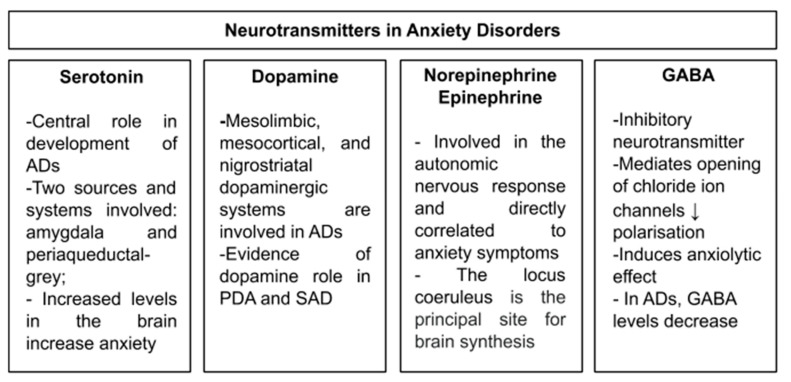
Neurotransmitters involved in anxiety disorders investigated in the present article. Note: ADs: anxiety disorders; GABA: gamma-aminobutyric acid; PDA: panic disorders with or without agoraphobia; SAD: social anxiety disorder.

**Figure 2 brainsci-10-00564-f002:**
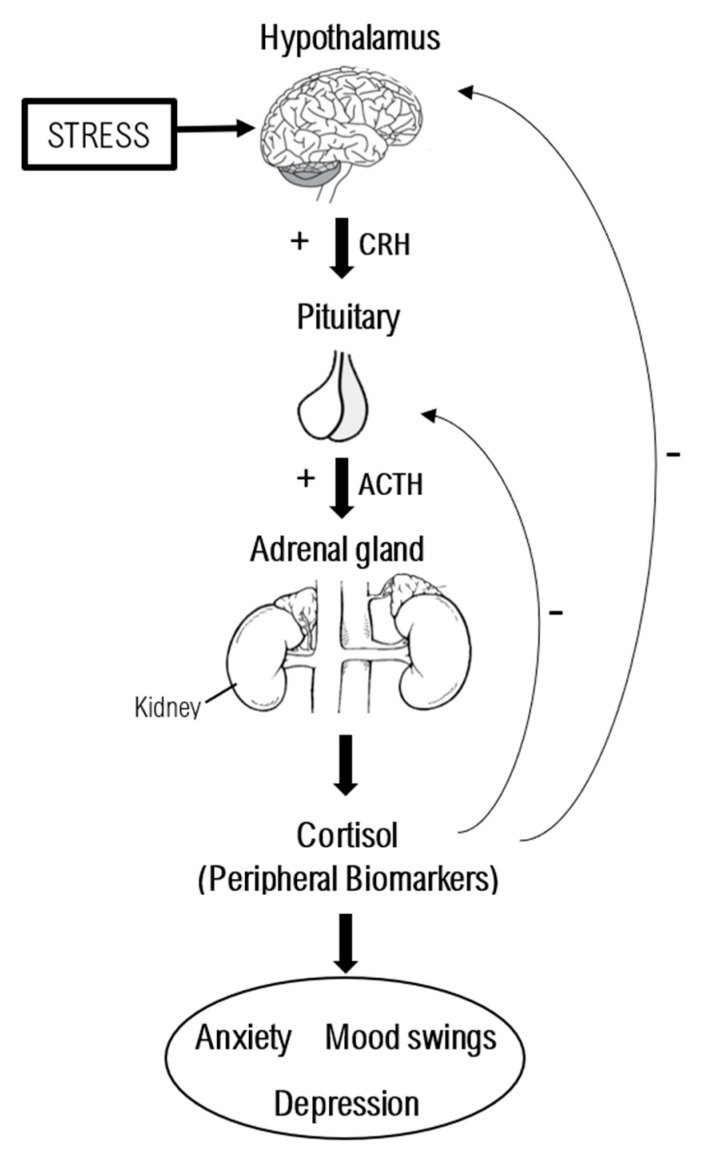
Illustration of the hypothalamic–pituitary–adrenal axis and related peripheral biomarkers. The hypothalamic–pituitary–adrenal (HPA) axis is an endocrine system based on feedback interactions among the hypothalamus, the anterior pituitary gland, and the adrenal glands. The neuroendocrine cells of the hypothalamus produce CRH (corticotropin-releasing hormone), which is then released into the adenohypophysis where it induces the synthesis of ACTH (adrenocorticotropic hormone). Finally, the adrenal cortex produces glucocorticoids (mainly cortisol) in response to ACTH stimulation. The stress-related dysregulation of the HPA axis in mood and anxiety disorders can be assessed by measuring the peripheral biomarkers, such as plasma, urinary, hair, and salivary cortisol, or the releasing factors (CRH and ACTH) and neurotransmitters (epinephrine and norepinephrine).

**Table 1 brainsci-10-00564-t001:** Summary of the literature findings related to the neurotransmitters involved in general anxiety disorders (GAD), panic disorders with or without agoraphobia (PDA), and social anxiety disorder (SAD).

	Biomarker	Finding
**General anxiety disorder**		
**Serotoninergic system**	Platelet 5-HT reuptake binding density	↓ vs. HCs [23] =vs. GAD, PDA, and HCs [24]
	Lymphocytes 5-HT reuptake binding density	=vs. HCs [25]
	5-HT and 5-HIAA concentration in platelet-rich and -poor plasma and in lymphocytes	=vs. HCs [25]
**Noradrenergic system**	Platelet alpha-2 adrenergic peripheral receptor binding density	↓ vs. HCs [26,27]
**GABAergic system**	Platelet peripheral BDZ binding sites number	↓ vs. HCs [28]
	Lymphocytes peripheral BDZ binding sites number	↓ vs. HCs [29,30]
	BDZ peripheral binding sites number after treatment with BDZ	↑ vs. HCs [28,30]
**Panic disorders with or without agoraphobia**		
**Serotoninergic system**	5-HT plasma levels	↓ vs. HCs [24,31,32]
	Platelet 5-HT concentration	=vs. HCs [33,34]
	Platelet aggregation in response to 5-HT	↓ vs. HCs [35]
	Platelet 5-HT uptake	↓ vs. HCs [23,35,36,37] =vs. HCs [24,38,39,40,41,42,43] ↑ vs. HCs [42,44]
	5-HIAA CSF levels	=vs. HCs [45] ↓ in patients with positive response to TCAs [45] ↑ in MDD females with comorbid PDA vs. patients without comorbid PDA and vs. HCs [46]
	5-HIAA jugular venous overflow	↑ vs. HCs [47]
	Plasma anti-5-HT and 5-HT anti-idiotypic antibodies	↑ vs. HCs [48]
**Dopaminergic system**	D plasma level	↑ vs. HCs [49]
	GH levels in response to apomorphine	↑ vs. MDD [50]
	HVA CSF concentration	=vs. HCs [45]
**Noradrenergic system**	E plasma levels	↑ vs. HCs [49]
	Platelet aggregation in response to NE, platelet α-2-receptor density, and lymphocyte β-receptor density before and after TCAs treatment	↑ vs. HCs [35]
	Tritiated clonidine binding (to measure platelet α-2-adrenoceptors)	↓ vs. HCs [51]
	Standing plasma NE	↑ vs. HCs [51]
	Beta-2-adrenoceptor density in both lying and standing position	↑ vs. HCs [52]
	Plasma MHPG in alcoholics with PDA	↑ vs. alcoholics without PDA [53]
**GABAergic system**	BDZ receptor sensitivity (measured by the severity of panic attacks after flumazenil intravenous injection)	↑ vs. HCs [54]
	GABA agonistic 3α-reduced neurosteroids	↑vs. HCs [55] ↓ vs. HCs [56]
	GABA antagonistic 3α-reduced isomer	↑ vs. HCs [56]
	Platelets peripheral BDZ receptors	↓ vs. HCs and vs. OCD [57]
**Social anxiety disorder**		
**Serotoninergic system**	Platelet 5-HT2 receptor density	=vs. HCs [58]
	Prolactin response to mCPP	=vs. HCs [59]
	Cortisol response to acute fenfluramine administration	↑ vs. HCs [60]
**Dopaminergic system**	D3 receptor desensitization after SSRIs treatment (measured by pramipexole impact)	↑ vs. HCs [61]

Note: 5-HT: serotonin; 5-HIAA: 5-hydroxyindoleacetic acid; BDZ: benzodiazepine; CSF: cerebrospinal fluid; D: dopamine; E: epinephrine; GABA: gamma-aminobutyric acid; GAD: generalized anxiety disorder; GH: growth hormone; HCs: healthy controls; HVA: homovanillic acid; mCPP: meta-chlorophenylpiperazine; MDD: major depressive disorder; MHPG: 3-methoxy-4-hydroxyphenylglycol; NE: norepinephrine; OCD: obsessive compulsive disorder; PDA: panic disorder with or without agoraphobia; SAD: social anxiety disorder; SSRIs: selective serotonin reuptake inhibitors; TCAs: tricyclics; ↑: increased; ↓: decreased.

**Table 2 brainsci-10-00564-t002:** Summary of the literature findings related to the neuropeptides involved in GAD, PDA, and SAD.

	Biomarker	Finding
**General anxiety disorder**		
**CCK**	Pentagastrin induced panic attacks rate	↑ vs. HCs [97]
**PACAP**	PACAP concentrations in females	↓ vs. HCs [86]
**Ghrelin**	Ghrelin plasma levels in children	↑ vs. HCs [98]
**Leptin**	Leptin circulating levels	=vs. GAD, SAD, and HCs [98]
**Panic disorders with or without agoraphobia**		
**CCK**	CCK-4 induced panic attacks rate	↑ vs. HCs [78]
	CCK-8 levels in CSF	↓ vs. HCs [99]
	CCK-8 levels in peripheral lymphocytes	↓ vs. HCs [100]
**Leptin**	Leptin plasma levels	=vs. PDA and HCs [101]
	Leptin serum levels	correlated with disease severity in female patients [102]
**ANP**	ANP baseline plasma levels	↓ vs. HCs [103]
	ANP plasma levels after lactate induced panic attacks	↑ vs. HCs [104]
**Adiponectin**	Adiponectin plasma levels	↓ vs. HCs [105]
		=vs. PDA and HCs [101]
**Social anxiety disorder**		
**OXT**	OXT plasma levels	=vs. SAD and HCs [106]
		correlated with disease severity [107]

Note: ADs: anxiety disorders; ANP: atrial natriuretic peptide; CCK: cholecystokinin; CSF: cerebrospinal fluid; GAD: generalized anxiety disorder; HCs: healthy controls; OXT: oxytocin; PACAP: pituitary adenylate activator polypeptide; PDA: panic disorder with or without agoraphobia; SAD: social anxiety disorder; ↑: increased; ↓: decreased.

**Table 3 brainsci-10-00564-t003:** Summary of the findings related to the hypothalamic–pituitary–adrenal axis involved in GAD, PDA, and SAD.

	Biomarker	Finding
**General anxiety disorder**		
**Basal levels**	Plasma ACTH concentration	↑ vs. HCs [140]
	Plasma cortisol level	=vs. HCs [140,141]
		↑ vs. HCs [142]
	Salivary cortisol levels	=vs. HCs [107,143]
		↑ vs. HCs and positively correlated with symptoms [144]
	Hair cortisol levels	↓ vs. HCs [145,146]
	sAA/cortisol ratio	↑ vs. HCs [147]
	Salivary cortisol awakening response	↑ in GAD with comorbid MDD vs. HCs [148]
		↓ vs. HCs in adults (>65 years) with long-lasting ADs [149]
	Plasma DHEAS levels	=vs. HCs [141]
	Cortisol/ DHEAS ratio	=vs. HCs [141]
**After stress-tests/treatments**	Plasma cortisol levels (after stress-test)	=vs. HCs [140]
	Plasma ACTH concentration (after stress-test)	=vs. HCs [140]
	Salivary cortisol levels (after a common stressful situation)	In males, ↓ decline rate of post-stress cortisol increases vs. HCs [143]
		In females, ↑ post-stress cortisol secretion and ↓ recuperation capacity correlated with clinical severity [143]
	Plasma cortisol levels (following CBT, escitalopram, and treatments refocusing patients’ attention)	↓ vs. HCs [150,151,152]
	Plasma cortisol levels (after treatment with buspirone, alprazolam, or diazepam)	=pre vs. post treatment [153,154,155,156]
	sAA/cortisol ratio (after mental arithmetic challenge)	↓ vs. HCs [147]
	HPA axis modifications (after dexamethasone suppression test)	=vs. MDD [142,157,158,159,160]
**Panic disorders with or without agoraphobia**		
**Basal levels**	Cortisol plasma levels	↑ vs. HCs [161,162,163,164]
		=vs. HCs [165,166]
	Urinary free cortisol levels	=vs. HCs [167]
		↑ vs. HCs [168]
		↑ in complicated PDA vs. HCs [169]
		=in active (paroxetine) vs. placebo treatment [170]
	Plasma ACTH concentration	↑ vs. HCs [166]
**After stress-tests/treatments**	Plasma ACTH concentration (response to CRH)	↓ vs. HCs [162,166,171]
		= vs. HCs [172]
	Plasma cortisol levels (response to CRH)	↓ vs. HCs [162,166]
		=vs. HCs [171,172]
	Plasma cortisol levels (after spontaneous panic attacks)	=vs. HCs [173]
	Plasma cortisol levels (after feared situations)	=vs. HCs [174]
	Plasma cortisol levels (during panic attacks induced by lactate infusion)	=vs. HCs [175,176,177,178,179,180]
		↑ vs. HCs [181]
		↓ vs. HCs [59]
	Plasma cortisol levels (after yohimbine-induced panic attacks)	↑ vs. HCs [182]
	Plasma cortisol levels (after caffeine or mCPP administration)	=vs. HCs [183,184]
	Salivary cortisol levels (during spontaneous panic attacks)	↑ vs. after [185]
	Plasma ACTH concentrations (after feared situations)	=vs. HCs [174]
	Plasma ACTH concentrations (during panic attacks induced by lactate infusion)	=vs. HCs [175,176,177,178,179,180]
**Social anxiety disorder**		
**Basal levels**	Urinary free cortisol levels	=vs. HCs [186,187]
	Salivary cortisol levels	=vs. HCs [188,189,190]
		=in female adolescents vs. HCs [191]
		risk factor for SAD onset [192]
	Salivary cortisol awakening response	↑ in SAD patients with comorbid MDD [148]
		↑ in children vs. HCs and significantly predicted SAD onset [193]
	Plasma cortisol levels	=vs. HCs [188]
	sAA/cortisol ratio	↑ vs. HCs [189,190]
**After stress-tests/treatments**	Salivary cortisol levels (after TSST)	=vs. HCs [188]
	Salivary cortisol levels (after dexamethasone)	=vs. HCs [187]
	Salivary cortisol levels (after intravenous citalopram, CCK-4 or mCPP)	=vs. HCs [194,195,196]
	Salivary cortisol levels (after administration of fenfluramine or mCPP)	↑ vs. HCs [59,197]
	Salivary cortisol levels (after stress test)	↑ vs. HCs [198,199]
	Plasma cortisol levels (after TSST)	=vs. HCs [188,200]
		↑ in SAD patients with comorbid MDD vs. HCs [200]
	sAA/cortisol ratio (after TSST)	=vs. HCs [188]
	sAA/cortisol ratio (after electrical stimulation)	↑ vs. HCs [190]
	sAA/cortisol ratio (after dexamethasone)	↑ vs. HCs [189]
	sAA/cortisol ratio (after public speaking task)	↑ vs. HCs [201]
	Hair cortisol concentration (after TSST)	=vs. HCs [188]
	Plasma ACTH concentration (after TSST)	↑ in SAD patients with MDD vs. HCs [200]
		=vs. HCs [200]

Note: ACTH: adrenocorticotropic hormone; ADs: Anxiety disorders; CBT: cognitive-behavioral therapy; CRH: corticotropin-releasing hormone; DHEAS: dehydroepiandrosterone; DST: dexamethasone suppression test; GAD: generalized anxiety disorder; HCs: healthy controls; MDD: major depressive disorder; mCPP: meta-chlorophenylpiperazine; PDA: panic disorder with or without agoraphobia; sAA: salivary alpha-amylase; SAD: social anxiety disorder; TSST: trier social stress test; ↑: increased; ↓: decreased.

**Table 4 brainsci-10-00564-t004:** Summary of the findings related to the neurotropic factors involved in GAD, PDA, and SAD.

	Biomarker	Finding
**General anxiety disorder**		
**BDNF**	Plasma levels	=vs. HCs, ↓ in females [220,221]
		No association with disorder severity [222]
		↓ vs. HCs [210,223,224]
		↑ vs. HCs, with normalization after remission [225]
	BDNF levels in the newborn’s cord blood of women with GAD	↑ vs. HCs [226]
**NGF**	Serum levels	=vs. HCs at baseline, ↑ after remission [227]
**GDNF**	Serum BDNF/GDNF levels	↓ vs. HCs [228]
**Panic disorders with or without agoraphobia**		
**BDNF**	Plasma levels	=vs. HCs [101]
		=vs. HCs, ↓ in females [221]
		↓ in patients with a greater risk to develop panic symptoms [218]
		↓ in patients with inadequate treatment response to CBT [229]
**NGF**	Serum levels	=vs. HCs [223]
**GDNF**	Serum levels	↑ vs. HCs [213]
**Social anxiety disorder**		
**BDNF**	Plasma levels	=vs. HCs, ↓ in females [221]
**GDNF**	Serum levels	↑ vs. HCs [213]

Note: BDNF: brain derived neurotrophic factor; CBT: cognitive behavioral therapy; GAD: generalized anxiety disorder; GDNF: glial cell line-derived neurotrophic factor; HCs: healthy controls; NGF: nerve growth factor; PDA: panic disorder with or without agoraphobia; NGF: nerve growth factor; SAD: social anxiety disorder; ↑: increased; ↓: decreased.

**Table 5 brainsci-10-00564-t005:** Summary of the findings related to the inflammation and immune system involved in GAD, PDA, and SAD.

	Biomarker	Finding
**General anxiety disorder**		
**Cytokines**	Peripheral levels of CRP, INF-γ, TNF-α, IL-1, IL-1α, IL-2, IL-6, IL-8, IL-10, IL-12p70, MCP-1, SDF-1, CM-CSF	↑ vs. HCs [245]
**Oxidative system**	Total oxidant status and oxidative stress index	↑ vs. HCs [246,247]
	Nitro-oxidative stress	↑ vs. HCs [248]
	Peripheral superoxide dismutase and catalase concentration	↑ vs. HCs [248]
		↓ vs. HCs [249]
	Paraoxonase-1 serum levels	↓ vs. HCs [250]
		↑ vs. HCs [248,251]
	Free sulfhydryl groups levels	↓ vs. HCs [250]
	Lipid hydroperoxides levels	↑ vs. HCs [248,251]
	Lipid-associated antioxidant defenses levels and HDL-cholesterol levels	↓vs. HCs [248]
	Malondialdehyde levels	↑ vs. HCs [249]
	Uric acid levels	↑ vs. HCs [248]
**Panic disorders with or without agoraphobia**		
**Cytokines**	Lymphocyte proliferative response to phytohemagglutinin and IL-2 production	↓ vs. HCs [252]
		=vs. HCs after treatment [252]
	Serum IL-6, IL-1β and IL-5 levels	↑ vs. HCs [253]
	IL-6 peak reaction after a psychosocial stress test	correlated with disease severity [254]
	Serum IL-10 levels	↑ vs. HCs [254]
	Elevated KYN/tryptophan ratio	correlated with poor short-term verbal memory [255]
	C3a, C5a, and C5b levels before and after pharmacological treatment	↓ vs. HCs [256]
	MBL and MBL-associated serine protease-2 levels	↓ vs. HCs [257]
	Peripheral lymphocyte subsets	=vs. HCs [258]
	CD3+, CD4+, and CD8+ T lymphocyte proportion after SSRIs treatment	↑ vs. HCs [258]
	B lymphocyte proportion after SSRIs treatment	↓ vs. HCs [258]
	CD19+ B lymphocytes number	↓ vs. HCs [259]
	CD4+ cells number	↓ vs. HCs [260]
	ADA and DPPIV serum levels	↑ vs. HCs [252]
**Social anxiety disorder**		
**Cytokines**	CRP and IL-6 plasma levels	↓ in females vs. HCs [236]

Note: ADA: adenosine deaminase; CRP: c-reactive protein; DPPIV: dipeptidyl peptidase IV; GAD: generalized anxiety disorder; GM-CSF: granulocyte-macrophage colony-stimulating factor; HCs: healthy controls; HDL: high-density lipoprotein; IL: interleukin; INF-γ: Interferon gamma; KYN: kynurenine; MBL: mannose-binding lectin; MCP-1: monocyte chemoattractant protein-1; PDA: panic disorder with or without agoraphobia; SAD: social anxiety disorder; SDF-1: stromal derived factor-1; SSRIs: selective serotonin reuptake inhibitors; TNF-α: tumor necrosis factor alpha; ↑: increased; ↓: decreased.

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
