# Peer review of "Peripheral Biomarkers in DSM-5 Anxiety Disorders: An Updated Overview"

_brainsci, 2020, doi:10.3390/brainsci10080564_

Round 1

Reviewer 1 Report

The manuscript submitted by Vismara and colleagues provides a very detailed overview of peripheral blood biomarkers in anxiety disorders. The manuscript is well written and falls within the scope of the journal. Main main concern is that the literature search strategy is not explained at all. Authors should provide a detailed protocol that includes the databases used, the words used in the literature search, exclusion criteria etc. Ideally, authors should include a flow-chart as suggested by the PRISMA-statement: http://prisma-statement.org/PRISMAStatement/FlowDiagram.

A further aspect that needs to be improved is the presentation and discussion of the finding of markers in different types of anxiety disorders (general anxiety disorders, panic disorders, social anxiety disorder). Instead of just summarizing the findings authors should clearly point out similarities and differences among these different subtypes of anxiety disordres and attempt an explanation for those differences/similarities. Authors should also discuss the potential of these biomarkers to improve diagnosis of anxiety disorders and their subcategories and whether they might offer predictive value.

A clear definition of the peripheral biomarkers under investigation should be given. Besides the reported blood-based biomarkers many other biomarkers can be measured in the blood (e.g. phosphatidylcholines, sphingomyelines, acylcarnitines, amino acids, sugars, etc.), which are also related to inflammation for instance. Please specify.

Author Response

Reviewer n° 1

1. “Main concern is that the literature search strategy is not explained at all. Authors should provide a detailed protocol that includes the databases used, the words used in the literature search, exclusion criteria etc. Ideally, authors should include a flow-chart as suggested by the PRISMA-statement: http://prisma-statement.org/PRISMAStatement/FlowDiagram.”

We thank the Reviewer for the comment. Our article aimed to provide a narrative review on the peripheral biomarkers in DSM-5 anxiety disorders and not a systematic review. Therefore, we did not follow PRISMA-statement criteria nor a quality assessment of selected articles.

In the Introduction section of the revised manuscript, we briefly described the methodology of our literature search strategy, as follows: “For the purpose of the present study, a literature search was conducted on the PubMed database considering articles published up until April 2020. Key search queries included combinations of the following terms: “biomarkers”, “anxiety disorders”, “peripheral biomarker”, “blood biomarker”, “generalized anxiety disorder”, “panic disorder”, “social anxiety disorder”, and “psychiatry”, In addition, reference lists of selected articles were screened for additional research. Authors included articles published in English that focused on the most investigated peripheral biomarkers related to neurotransmitters, neuropeptides, the HPA axis, neurotrophic factors, and inflammation and immune system”.

Recognising the possible biases of a narrative review, we addressed this issue in the limitation section, as follows: “Finally, considering the narrative nature of the present review article, possible other literature articles might have been overlooked”.

2.“A further aspect that needs to be improved is the presentation and discussion of the finding of markers in different types of anxiety disorders (general anxiety disorders, panic disorders, social anxiety disorder). Instead of just summarizing the findings authors should clearly point out similarities and differences among these different subtypes of anxiety disorders and attempt an explanation for those differences/similarities. Authors should also discuss the potential of these biomarkers to improve diagnosis of anxiety disorders and their subcategories and whether they might offer predictive value”.

We thank the Reviewer for the comment. Through all the article and particularly in the Conclusion section, we further implemented the discussion of the findings related to the investigated biomarkers, trying to point out similarities and differences among the anxiety disorders and how the selected biomarkers might help in improving the diagnosis and treatment of these disorders. Please see the track-changed version of the manuscript for the edits and, in particular, the Conclusion section, that now states: “Overall, despite promising results, most reports have produced solitary findings, sometimes inconsistent and not clearly replicable and applicable to clinical practice. In fact, even though these studies have provided a relevant contribution to increase the knowledge of neurobiological mechanisms of ADs, no specific dysfunction of a particular neurotransmitter or a neuropeptide could be defined as the main cause for ADs and, consequently, be chosen as a diagnostic peripheral biomarker. Among other reasons, this might be related to the limited sample size of most studies in the field and to the influence of some clinical variables (i.e., gender, age, concomitant medication, comorbidities, clinical severity) as well as to study methodology. Moreover, in some cases, the same biomarker showed to play a role also in other psychiatric disorders than ADs (e.g. BDNF in depression [269] or OCD [270], OXT in depression or schizophrenia [271]), emphasising how psychiatric disorders exist in a continuum expression of symptomatology and common genes and pathological mechanisms are underpinned.

In summary, biomarkers related to GAD showed some degree of consistency with respect to the GABAergic system, with most of the studies reporting a reduction of GABA-related biomarkers in GAD patients [30–32]. Although the HPA axis is dysregulated also in GAD [138], no consistent results can be drawn with respect to peripheral biomarkers, with some studies showing an increase, while others a decrease of investigated biomarkers, at baseline or after stress test/treatments. BDNF showed some consistent results (reduction of basal levels in some reports [210,220,225,226]), likely supporting its role in different ADs and encouraging further investigation [218]. In PDA, several studies showed an increase of peripheral biomarkers related to the noradrenergic system [37,53–55,59], underling an abnormal noradrenergic system regulation with consequent pathological clinical expression. Among neuropeptides, several findings are related to abnormalities in the CCK system in PDA patients [78,119,120], and potential new molecules (CCK-receptor antagonists) targeting this system have been investigated [99,102]. For SAD, findings related to peripheral biomarkers are mostly solitary, probably due to the relative minor epidemiologic and clinical impact of this disorder, therefore no peripheral biomarker of diagnostic value can be chosen”.

Moreover, all peripheral biomarkers of treatment response remain in very early stages of development and only a few field studies have demonstrated reliability for predicting pharmacological response. Results from the present article showed that some peripheral biomarkers related to neurotransmitters were associated with treatment response. Among these, CFS 5-HIAA level in GAD patients emerged to be significantly decreased in patients with a positive response to tricyclics [47]. In addition, a low pre-treatment β-adrenoceptor affinity was associated with a positive response to paroxetine [56]. With respect to the HPA axis in GAD, a reduction of cortisol levels was associated with a positive response to CBT [156], escitalopram [157], and treatments refocusing patients’ attention [158]. Some neurotropic factors showed a predictive value in treatment response. In GAD patients, a reduction of BDNF levels followed symptoms remission (after treatment with paroxetine) [226] and NGF serum levels, after successful CBT, increased significantly [230]. Moreover, in PDA, a lower serum BDNF level was associated with inadequate treatment response to CBT [231]. With respect to the inflammation and immune system, PDA patients showed a significantly increase of CD3+, CD4+, and CD8+ T lymphocyte proportions, and a decreased B lymphocyte proportion after effective paroxetine treatment and, in the same disorder, a change in the lymphocyte proliferative response to phytohemagglutinin correlated with a reduction in self-reported anxiety level after treatment [260]. Although the abovementioned results mostly represent solitary findings, they constitute the first step for additional studies targeting these peripheral biomarkers.

3. “A clear definition of the peripheral biomarkers under investigation should be given. Besides the reported blood-based biomarkers many other biomarkers can be measured in the blood (e.g. phosphatidylcholines, sphingomyelins, acylcarnitines, amino acids, sugars, etc.), which are also related to inflammation for instance. Please specify”.

We thank the Reviewer for the comment. The available literature about peripheral biomarkers in psychiatric disorders is extensive and different molecules might be considered potential biomarkers for these disorders. Indeed, although some definitions of biomarkers are more frequently accepted, no univocal definition is present in the literature.

In our review, we decided to focus on the most investigated biomarkers that are related to neurotransmitters, neuropeptides, the HPA axis, neurotrophic factors, and inflammation and immune system, considering these systems as the ones traditionally more implicated in the etio-pathogenesis of anxiety disorders. Among these systems, different molecules could be taken into consideration, but we have decided to focus primarily on the ones which have been more extensively investigated in the literature. Moreover, we reported in our paper literature articles that specifically focused on ADs (GAD, PDA, or SAD) and reported a comparison with a control condition (i.e., healthy controls or other psychiatric disorders).

To clarify this issue, we specified in the Introduction section the following: “Authors included articles published in English that focused on the most investigated peripheral biomarkers related to neurotransmitters, neuropeptides, the HPA axis, neurotrophic factors, and inflammation and immune system”.

Moreover, we specified in the Conclusion section that other molecules might have been considered in other reports (not included in our article) as potential biomarkers for ADs, as follows: “Moreover, in the present article the authors focused on the most investigated peripheral biomarkers in ADs; therefore, other molecules not herein mentioned might have been considered as biomarkers in other investigations”.

Reviewer 2 Report

The current field of personalized/precision medicine seeks to use multidimensional approaches including genetic and epidemiological research to tailor therapeutics, prevent disease, and promote health. In order to further the development of precision medicine, identification of diagnostic, prognostic and treatment biomarkers for different diseases is an important step. While promising biomarkers have emerged in the field of cancer and cardiovascular medicine, the field of psychiatry still suffers from lack of reliable biomarkers. In contrast to other disciplines, the use of biomarkers in psychiatry is associated with specific challenges the most important being limited accessibility to the central nervous system in living patients. This is one of the reasons why extensive research has focused on peripheral biomarkers. The current and timely review by Vismara et al is aimed at providing an upto date status of putative peripheral biomarkers in anxiety disorders. This is also important since in the last edition of the DSM no biomarkers were reported, though their inclusion was a goal originally aspired.

The authors have included studies that have investigated neurotransmitters, neuropeptides, markers of the HPA axis, neurotrophic factors and inflammatory markers in blood/CSF of patients with generalized anxiety disorder, panic disorder with agoraphobia and social anxiety disorder. The authors conclude that while there are some promising results, most reported investigations have revealed solitary findings, mostly inconsistent and therefore currently not applicable in clinical practice.

As mentioned before, the topic chosen here is very important. However, there are major pitfalls in the way the review is written, as outlined below.

Major points

  1. The authors have taken into account studies investigating diagnostic/prognostic biomarkers in the blood in anxiety disorders. Only in a few instances the authors have included biomarker studies in response to therapeutic treatments, eg. in table 2: ‘’treatment with fluvoxamine reduces the sensitivity of PDA patients to CCK-4’’. The authors should make sure to provide a complete picture concerning (including responder and non-responders). Although current treatments often are not optimal and therefore might not fully validate the diagnostic/prognostic marker, this information still is important and needs to be included.
  2. In order to cater to a broader audience, a figure/scheme/diagram showing different components of blood, receptor bindings sites within specific blood subcompartments, and other necessary details (wherever applicable) should be provided and a general introductory section in the text referring to this overview should be presented.
  3. The tables need to be rearranged. I have presented a format below as an example. Also instead of increased or decreased, symbols ↑ or ↓ could be respectively used in the table to enhance readability. Finally, the authors should not exclusively present biased ‘positive’ findings wherein changes have been observed but rather provide a realistic, unbiased picture of the current evidence. Some nice examples of tables can be found in (Schiele et al, 2020, Clin Psychol Rev.; Hammamieh et al, 2017, Trans. Psy. )

Example:

Generalized anxiety disorders

Baseline

(peripheral biomarker)

         Treatment outcome

(changes in anxiety symptoms)

Treatment outcome

(changes in peripheral biomarker)

Neurotransmitter

Serotonin system

Platelet study 1

Platelet study 2

Platelet study 3

Plasma study 1

Plasma study 2

Plasma study 3

Platelet study 1

Platelet study 2

Platelet study 3

Plasma study 1

Plasma study 2

Plasma study 3

Platelet study 1

Platelet study 2

Platelet study 3

Plasma study 1

Plasma study 2

Plasma study 3

Dopamine system

Noradrenergic system

Neuropeptide

CCK

PACAP

  1. Recently, the National Institutes of Health has recognized the gender gap in scientific knowledge in scientific research and now mandates that studies be conducted in both sexes and to include gender as variables influencing physiological processes (http://orwh.od.nih.gov/sexinscience/overview/pdf/NOT-OD-15-102_Guidance.pdf). Among the studies included by the authors, there are many studies wherein gender differences are reported for eg. ref. 98. However, the authors have not taken these differences into account. Please include this aspect (also in table section).
  2. In a majority of the sections such as line 149-156; 253-256; 257-260; 293-296; 524-529 etc. the authors have simply stated the published findings. The written text does not offer any additional information than the literature in the table. Interpretation and Authors’ scientific view behind the findings should be appropriately presented.
  3. In the conclusion section, the future perspective of such biomarker research is missing. Since the authors conclude that the current findings are not applicable in clinical practice, please provide suggestion how to proceed, eg. how to improve the reliability of peripheral biomarkers in the anxiety field.

Author Response

Reviewer n°2

1. The authors have taken into account studies investigating diagnostic/prognostic biomarkers in the blood in anxiety disorders. Only in a few instances the authors have included biomarker studies in response to therapeutic treatments, eg. in table 2: ‘’treatment with fluvoxamine reduces the sensitivity of PDA patients to CCK-4’’. The authors should make sure to provide a complete picture concerning (including responder and non-responders). Although current treatments often are not optimal and therefore might not fully validate the diagnostic/prognostic marker, this information still is important and needs to be included.

We thank the Reviewer for the comment.  We had to bear in mind that most of the selected studies were not specifically therapeutic trials. Consequently, some outcome variables (e.g. response rates) were not reported or were not the main aim of the investigation. As requested by the Reviewer, we better specified in the revised manuscript additional variables related to therapeutic treatments for selected studies that investigated biomarkers’ role in relation to therapeutic strategies. Please see the edited version of the manuscript for new changes throughout the text.

2. In order to cater to a broader audience, a figure/scheme/diagram showing different components of blood, receptor bindings sites within specific blood subcompartments, and other necessary details (wherever applicable) should be provided and a general introductory section in the text referring to this overview should be presented.

We thank the Reviewer for the suggestion and we also believe that catering a broader audience will increase readability and diffusion of the article. Without forgetting manuscript guidelines – particularly in terms of article extension, we included some short introduction sentences to better explain and present the investigated topics (in particular to the Neuropeptides section). Moreover, we included two new figures: Figure 1, showing the neurotransmitters involved in ADs that were investigated in the present paper, and Figure 2, picturing the HPA axis function and the peripheral biomarkers involved.

3. The tables need to be rearranged. I have presented a format below as an example. Also instead of increased or decreased, symbols ↑ or ↓ could be respectively used in the table to enhance readability. Finally, the authors should not exclusively present biased ‘positive’ findings wherein changes have been observed but rather provide a realistic, unbiased picture of the current evidence. Some nice examples of tables can be found in (Schiele et al, 2020, Clin Psychol Rev.; Hammamieh et al, 2017, Trans. Psy. )

We thank the Reviewer for the suggestion. According to Reviewer’s suggestion, all tables were rearranged. We thank the Reviewer for the proposed format as an example: however, we adopted a different structure that, nonetheless, focuses on investigated biomarkers. Moreover, we kept table topics sorted by systems involved in order to place tables after the referring section. Lastly, new Tables include all positive and negative findings and symbols ↑ or ↓ have been used to enhance readability, as suggested.

4. “Recently, the National Institutes of Health has recognized the gender gap in scientific knowledge in scientific research and now mandates that studies be conducted in both sexes and to include gender as variables influencing physiological processes (http://orwh.od.nih.gov/sexinscience/overview/pdf/NOT-OD-15-102_Guidance.pdf). Among the studies included by the authors, there are many studies wherein gender differences are reported for eg. ref. 98. However, the authors have not taken these differences into account. Please include this aspect (also in table section)”.

We thank the Reviewer for this important suggestion. In our article, some cited studies included only male or female patients (ref n° 48, 133, 140, 193) and this sample characteristic has been specified accordingly.

In other cited articles, showing gender-related differences, we further specified the role of gender (ref n° 86, 108, 144, 220, 222, 226, 236, 241). Additionally, we further reassessed all the articles included in order to clarify the potential role of gender. Please see the track-changed version of the manuscript for edits.

5. “In a majority of the sections such as line 149-156; 253-256; 257-260; 293-296; 524-529 etc. the authors have simply stated the published findings. The written text does not offer any additional information than the literature in the table. Interpretation and Authors’ scientific view behind the findings should be appropriately presented”.

We thank the Reviewer for the comment. This issue has been already addressed following the other Reviewer’s suggestion. Due to the vast literature findings on this topic, often with mixed results, it was not always possible to draw direct conclusions. When this was not the case, we further discussed the results of selected articles. Please see the track-changed version of the manuscript for new edits in this specific regard.

6. In the conclusion section, the future perspective of such biomarker research is missing. Since the authors conclude that the current findings are not applicable in clinical practice, please provide suggestion how to proceed, e.g. how to improve the reliability of peripheral biomarkers in the anxiety field.

We thank the Reviewer for the comment. In the Conclusion section, we included Authors’ suggestions on how to implement applicability on clinical practice. Please see the Conclusion section where we stated: “Thus, determining a practical biomarker in patient-derived peripheral samples that helps in the correct diagnosis, prognosis, and prediction of response to treatment remains highly desirable. One of the first step to increase the identification of a potential biomarker is to better define which “problem” is investigated. To do so, scientific searches should narrow their focus and clearly define what a biomarker is aimed to investigate (e.g., which disease phenotype, clinical diagnostic group, gender, age). Following this purpose, possible biases related to studies’ methodology could be overtaken. Consequently, multicentric investigational studies should be encouraged in order to collect a large number of patients with narrow identical baseline characteristics that accurately represent the population at-large.

If the discovery of an ideal biomarker is achieved, its use in clinical practice should be further investigated and implemented, considering additional characteristics like costs, patients’ tolerability, and overall clinical applicability. To do so, further economic resources should be invested to implement the technologies that permit practical, affordable, and robust sampling and diagnostic techniques that can be routinely used in research and in clinical practice.

Lastly, a future direction might consist of adopting a machine learning approach, which has been previously used to address mental health questions [276]. Peripheral biomarkers, combined with other biomarkers (related to genetic, epigenetic, neuroimaging, and neurophysiology) and clinical variables, might contribute to the machine learning algorithms that could calculate the risk to develop a specific AD, determine the clinical evolution, and find precise and personalized treatments”.

Round 2

Reviewer 1 Report

The manuscript has improved considerably. I have no further comments.